# Comprehensive Metabolomics and Machine Learning Identify Profound Oxidative Stress and Inflammation Signatures in Hypertensive Patients with Obstructive Sleep Apnea

**DOI:** 10.3390/antiox11101946

**Published:** 2022-09-29

**Authors:** Zhiyong Du, Haili Sun, Yunhui Du, Linyi Li, Qianwen Lv, Huahui Yu, Fan Li, Yu Wang, Xiaolu Jiao, Chaowei Hu, Yanwen Qin

**Affiliations:** 1The Key Laboratory of Remodeling-Related Cardiovascular Diseases, Ministry of Education, National Clinical Research Center for Cardiovascular Diseases, Beijing Anzhen Hospital, Capital Medical University, Beijing 100029, China; 2Beijing Institute of Heart Lung and Blood Vessel Disease, Beijing 100029, China

**Keywords:** hypertension, obstructive sleep apnea, metabolomics, biomarkers, diagnosis, oxidative stress, inflammation

## Abstract

Obstructive sleep apnea (OSA) can aggravate blood pressure and increase the risk of cardiovascular diseases in hypertensive individuals, yet the underlying pathophysiological process is still incompletely understood. More importantly, OSA remains a significantly undiagnosed condition. In this study, a total of 559 hypertensive patients with and without OSA were included. Metabolome and lipidome-wide analyses were performed to explore the pathophysiological processes of hypertension comorbid OSA and derive potential biomarkers for diagnosing OSA in hypertensive subjects. Compared to non-OSA hypertensive patients (discovery set = 120; validation set = 116), patients with OSA (discovery set = 165; validation set = 158) demonstrated a unique sera metabolic phenotype dominated by abnormalities in biological processes of oxidative stress and inflammation. By integrating three machine learning algorithms, six discriminatory metabolites (including 5-hydroxyeicosatetraenoic acid, taurine, histidine, lysophosphatidic acid 16:0, lysophosphatidylcholine 18:0, and dihydrosphingosine) were selected for constructing diagnostic and classified model. Notably, the established multivariate-model could accurately identify OSA subjects. The corresponding area under the curve values and the correct classification rates were 0.995 and 96.8% for discovery sets, 0.997 and 99.1% for validation sets. This work updates the molecular insights of hypertension comorbid OSA and paves the way for the use of metabolomics for the diagnosis of OSA in hypertensive individuals.

## 1. Introduction

Hypertension is a leading modifiable risk factor for cardiovascular diseases, which represents the top cause of death worldwide [1,2]. Obstructive sleep apnea (OSA) is a very common respiratory disease caused by the collapse and obstruction of the upper airway during sleep [3]. More importantly, OSA frequently coexists with hypertension. Previous studies have found that 30–50% of hypertensive patients have OSA and around 50% of OSA patients may have hypertension [4,5]. Growing evidence suggests that OSA is associated with worse night-time blood pressure (BP) control, resistant hypertension development, and increased target organ damage in patients with hypertension [5,6,7].

Like hypertension, OSA also increases the risk of cardiovascular diseases [8]. When not treated, OSA is associated with an increased morbidity and mortality in a wide range of cardiovascular morbidities, such as stroke, atrial fibrillation, myocardial infarction, and heart failure [8,9,10]. Previous evidence indicated that patients with hypertension comorbid OSA were at an increased risk for developing cardiovascular diseases compared with their hypertensive counterparts without OSA [11,12]. Although recent studies have demonstrated that hypertension and OSA might have a synergistically negative effect on the cardiovascular system by causing sympathetic overactivity and inducing inflammation and oxidative stress [11,12,13], the underlying pathophysiological process is still incompletely understood.

It is worth noting that OSA is still undiagnosed and untreated in the majority of the hypertensive population [14]. The current gold standard for diagnosing OSA is overnight polysomnography (PSG), a time-consuming and expensive procedure, which hinders its utility in clinical diagnosis of OSA [15]. In the last decade, state-of-the-art metabolomic technologies have drawn great attention to OSA and hypertension. Metabolomics, the qualitative and quantitative analysis of metabolome and lipidome within biological fluids, aims to uncover the underlying pathophysiological processes of a biological system, and provides useful diagnostic or prognostic information [16].

Recent evidence revealed that dysregulation of metabolites or lipid molecules have been associated with BP and OSA severity, supporting the view of hypertension and OSA as metabolic diseases [17,18,19]. Furthermore, numerous studies have revealed that the metabolomic alterations might reflect the development and progression of cardiovascular disease [18,20]. Therefore, a comprehensive metabolomic examination in hypertensive patients with and without OSA can provide a better understanding of the potential pathophysiological effects of OSA comorbid hypertension on cardiovascular systems and offer personalized treatment strategies for the prevention of cardiovascular disease in hypertension populations. Furthermore, such knowledge can also potentially aid researchers and clinicians to improve diagnostic methods of OSA in a more convenient and precise way.

To date, a comprehensive metabolic profiling of hypertensive patients with and without OSA are still lacking. Herein, we utilized a combined metabolome and lipidome analysis to investigate the metabolic changes in a large panel of patients with OSA comorbid hypertension compared to non-OSA hypertensive patients. We also sought to explore the potential diagnostic value of metabolite signatures in identifying OSA from the hypertension population.

## 2. Materials and Methods

### 2.1. Subjects, Clinical Assessment, and Study Design

A total of 360 patients in Beijing Anzhen Hospital were enrolled between March 2017 and June 2018 as the discovery set and another 340 patients from January 2019 to November 2020 were included for the validation set. Inclusion criteria were: participants aged >18 years with definite hypertension. Exclusion criteria for the study were: current or historical diseases or conditions of all systems other than hypertension, mainly including serious cardiovascular and cerebrovascular diseases (myocardial infarction, stroke, heart failure, etc.), respiratory diseases (obstructive pulmonary disease, etc.), serious digestive diseases (ulcerative colitis, etc.), infectious diseases (HIV, Hepatitis B, etc.), chronic kidney diseases, pulmonary hypertension, pregnancy, malignancy, as well as known OSA with continuous positive airway pressure therapy. After screening (Figure 1), 559 individuals were included in the final analysis (discovery set = 285; validation set = 274). Verbal and written consent was obtained from all subjects. This study complies with the Declaration of Helsinki and was approved by the Ethics Committee of Beijing Anzhen Hospital of the Capital University of Medical Sciences.

BP parameters of the study subjects were recorded by using an office automatic and validated digital machine (Omron HEM-7124; Kyoto, Japan) and ambulatory blood pressure monitoring (Biox; VasoMedical, Westbury, NY, USA). Hypertension was defined as a condition in individuals with mean office systolic BP (SBP) ≥ 140 mmHg or diastolic BP (DBP) ≥ 90 mmHg (three consecutive measurements at 5-min intervals), which is equal to 24-h SBP ≥ 130 mmHg or 24-h DBP ≥ 80 mmHg or a condition in those who underwent antihypertensive agent therapy with a defined history of hypertension. Subjects with a nocturnal BP fall of less than 10% were classified as non-dippers and the others were classified as dippers [21]. Then, all the patients underwent a full overnight polysomnography study (Alice nightone; Respironics, Pittsburgh, PA, USA). OSA was defined as an apnea-hypopnea index (AHI) ≥ 15 events/hour of sleep. This cutoff relies on recent international recommendations [22] and the fact that evidence suggested that mild OSA (5 < AHI < 15) might not have significant cardiovascular consequences [23]. After the sleep study, the subjects were divided into two groups: OSA comorbid hypertension (discovery set = 165, validation set = 158) and hypertension without OSA (discovery set = 120, validation set = 116).

### 2.2. Data Collection and Clinical Laboratory Tests

Demographic characteristics including age, gender, body mass index (BMI), neck circumference, waist circumference, hip circumference, waist-to-hip ratio, heart rate, sleeping pills treatment, and antihypertensive drugs, smoking and drinking status were recorded for each study participant. Smoking status was defined as current smoker at the time of health checkup examination. Drinking status were defined as subjects with an alcohol consumption >10 g/day. Diabetes was defined as a condition in individuals with clinical diagnosed type 1 and type 2 diabetes based on the American Diabetes Association. Mean office BP, 24-h BP, day-time BP, night-time BP, AHI, lowest oxygen saturation (LSaO_2_), mean oxygen saturation (MSaO_2_), mean apnea-hypopnea duration (MAD), percentage of cumulative time with oxygen saturation below 90% (CT90), oxygen desaturation index (ODI), and arousal index (ArI) were also recorded. Sleepiness was evaluated using the Epworth sleepiness scale (ESS). Fasting blood samples were obtained to measure glucose, total cholesterol (TC), triglycerides (TG), low-density lipoprotein cholesterol (LDL-C), high-density lipoprotein cholesterol (HDL-C), non-HDL, alanine aminotransferase (ALT), and aspartate aminotransferase (AST) were determined using an automatic biochemistry analyzer (Beckman AU 5400, Brea, CA, USA).

### 2.3. Serum Metabolome and Lipidome Analyses

The blood sample was separated by centrifugation at 3000× *g* for 20 min. A total of 150 μL of supernatant serum was transferred into a new tube and mixed with 450 μL of ice-cold chloroform-methanol ((2:1, *v*/*v*) containing a series of isotope-labeled internal standards, including 0.005 mg/mL L-phenyl-*d*_5_-alanine, 0.004 mg/mL L-arginine-*d*_7_, 0.005 mg/mL L-valine-*d*_8_, 0.005 mg/mL stearic acid-18, 18, 18-*d*_3_, 0.008 mg/mL 4-methylpentanoic acid-*d*_12_, 0.01 mg/mL cholic acid-2, 2, 4, 4-*d*_4_, 0.011 mg/mL LysoPC (19:0)-*d*_5_, 0.013 mg/mL PC(18:0/20:4)-*d*_11_, 0.005 mg/mL (±) 15-HETE-*d*_8_, and 0.01 mg/mL stearoyl-L-carnitine-*d*_3_. The mixture was vortexed for 5 min at 4 °C and centrifuged at 14,500 rpm for 10 min at 4 °C. After precipitating the proteins, the upper aqueous phase and the lower organic phase were separately collected into two clean dry tubes for metabolomic and lipidomic analyses, respectively. Subsequently, all collected aqueous and organic phases were evaporated to dryness, and the dried residue was stored at −80 °C until further analysis. The measurements of serum metabolome and lipidome were followed as previously described [24,25]; the detailed LC-MS approaches are depicted Method in Supplementary Material.

### 2.4. Metabolic Data Processing and Pathway Analysis

All raw data were analyzed by using MS-DIAL software v3.6 for deconvolution, alignment, and data reduction to provide a comprehensive data matrix, including information of precursor ions, fragment ions, neutral molecules, retention times, and raw intensity. Metabolite identification was performed in MS-DIAL and Progenesis QI software (Waters, Manchester, UK) by comparing the exact molecular mass and fragments using QI MetaScope, HMDB, METLIN, and LIPIDMAPS databases. The matched metabolite was further identified by the isotopic distribution measurement and the retention times and MS/MS fragmentation patterns of authentic standards using in-house metabolite library. Then, the raw data matrix were normalized by the quality control (QC) samples and the intensity of internal standards. Metabolome and lipidome datasets were merged for subsequent statistical analysis.

Principal component analysis (PCA) was applied to gain a comprehensive view of samples’ distribution by using SIMCA-P software (v14.0, Umetrics, Umea, Sweden). The *S*-plot of partial least squares discriminate analysis (PLS-DA) and Mann Whitney *U* test (a false discovery rate (FDR)–adjusted *p* values < 0.05) was performed to identify the differentially expressed metabolites between groups. PLS-DA and orthogonal partial least square discriminant analysis (OPLS-DA) were used to investigate the potential effects of gender and antihypertensive drug treatment on the discriminatory metabolite signatures. The metabolic pathway enrichment analysis was performed by using MetaboAnalyst (http://www.metaboanalyst.ca/ (accessed on 6 June 2022)). The latent relationship network between functional pathways/diseases and metabolites was generated on the basis of Function Analysis, Connect Analysis, and Path Explorer by using Ingenuity Pathway Analysis (IPA, QIAGEN Inc., Hilden, German). Hierarchical clustering heat map analyses of metabolite features were performed by using TBtools software v1.082.

### 2.5. Important Metabolic Feature Identification and Diagnosis Model Performances

Biomarker discovery requires not only the optimization of the biomarker usefulness regarding the biological relevance, but also the number of biomarkers [26], to select a small number of the representative biomarkers that can also maintain a significant performance for OSA diagnosis in hypertensive individuals. The classification and feature ranking models were established by using three machine learning algorithms in MetaboAnalyst software [26,27,28], including PLS-DA, support vector machine (SVM), and random forest. In each classification model, twenty top metabolite biomarkers were selected on the basis of the average importance. The shared biomarkers identified from three classifiers were used to create biomarker models by using a logistic model-based receiver operating curve (ROC). The prediction and classification performances were evaluated by using the posterior classification probability (100 cross-validations) and permutation test (*n* = 500 times).

### 2.6. Statistical Analysis

Categorical variables were summarized by frequency (*n*) or percentages (%) and compared by using Chi-square test. Continuous and non-normally distributed variables were presented by mean and standard deviation (means ± SD) and medians and interquartile ranges [IQR], respectively. A two-tailed Student’s *t* test and Mann Whitney *U* test were used for the comparisons of normally distributed data and non-normally distributed data, respectively. A false discovery rate (FDR)-calibrated *p* < 0.05 was considered significant. The correlation of metabolites with OSA were performed by regression analyses using the sum of standardized metabolite values (z-scores) weighted according to the value of their corresponding β-coefficients by using SPSS Statistics 26 (IBM Corp, New York, NY, USA) and online bioinformatics platform (http://www.bioinformatics.com.cn (accessed on 28 June 2022)). Relationships between clinical/laboratory measures and metabolites were calculated using Spearman’s rank correlation coefficient and debiased sparse partial correlation (DSPC) network based on the online bioinformatics platform and MetaboAnalyst.

## 3. Results

### 3.1. Participant Clinical Characteristics

A total of 559 individuals participated in the study (Figure 1). In the discovery set, of the 285 hypertensive patients, 165 patients had OSA condition. In the validation set, 158 of 274 individuals had hypertension comorbid OSA. The clinical characteristics are summarized in Table 1. In both discovery and validation sets, no differences were observed in ages, gender, smoking and drinking status, the prevalence of diabetes mellitus, heart rate, blood levels of glucose, TC, LDL-C, non-HDL, AST, and ALT between hypertensive patients with OSA and patients without OSA. In contrast, hypertensive patients with OSA had higher values of neck circumference, waist circumference, hip circumference, waist-to-hip ratio, BMI, TG, and lower levels of HDL-C than patients without OSA. In BP measures, night-time SBP and DBP were significantly increased in the patients with hypertension comorbid OSA compared to those without OSA. Although no significant difference was observed in the prevalence of non-dippers in the comparisons of OSA and non-OSA subjects, the results demonstrate that patients with OSA (45.45% in discovery set; 48.73% in validation set) exhibited a higher prevalence trend than those without OSA (38.33% in discovery set; 41.38% in validation set). During the PSG monitoring, patients with hypertension comorbid OSA had poorer performance than those without OSA, as evidenced by significant alterations in AHI, ESS, LSaO_2_, MSaO_2_, MAD, CT90, ODI, and ArI.

### 3.2. Metabolic Phenotypes of Hypertensive Patients with and without OSA

Nontargeted profiling of metabolome and lipidome were employed to obtain the serum metabolic characteristics of the studied individuals as comprehensively as possible. The unsupervised principal component analysis (PCA) scores plot of all test samples and quality control (QC) samples as well as the relative standard derivations of the distribution for the identified metabolites in the QC samples are shown in Appendix A, and the results indicate that the present metabolic analyses were reliable. Subsequently, the 3D-PCA scores plots were generated for group separations. As shown in Figure 2A, a distinct discrimination between the sera profiles of patients with hypertension comorbid OSA and non-OSA hypertensive patients was observed in both discovery and validation sets. Additionally, the established PCA models were highlighted with satisfactory values of R2X and Q2 (discovery set: R2X = 0.643, Q2= 0.608; validation set: R2X = 0.565, Q2 = 0.509), indicating a good explanatory and predictive ability.

### 3.3. Identification of a Metabolic Fingerprint Specific for Hypertension Comorbid OSA

To characterize the differentiated metabolites in hypertensive patients with and without OSA, the *S*-plots of the supervised PLS-DA based on the discovery and validation datasets were constructed. As shown in Figure 2B, a variety of hydrophilic metabolites and lipid species showed significant contributions to the separation between hypertensive patients with OSA and non-OSA subjects. After further univariate non-parametric test confirmation, a total of 63 significantly differential metabolites with FDR-adjusted *p* < 0.05 were identified and are summarized in Appendix A. These metabolic alterations mainly included amino acid and its derivatives, organic acid and its derivatives, nucleotide and its derivates, amines, fatty acyls, diacylglycerol (DAG), lysophosphatidylcholine (LPC), and eicosanoids. The overall heat map of differential metabolites revealed that the samples of patients with hypertension comorbid OSA in the discovery and validation datasets clustered closely and are clearly separated from the samples of non-OSA hypertensive patients (Figure 3A).

Furthermore, the unsupervised and supervised multivariate statistical analyses of the metabolic alterations did not show clustering of samples by gender or antihypertensive drugs in both of the discovery and validation sets, indicating no discriminatory metabolite signatures due to gender or antihypertensive drugs of the study groups (Appendix A). Our results indicate that hypertensive patients with OSA exhibited several alterations in the clinical characteristics compared with those without OSA (Table 1). Importantly, several factors, such as BMI, BP, waist circumference, and waist-to-hip ratio are well-known metabolic regulators. Therefore, we next performed multivariate regression analysis to test whether the identified metabolic biomarkers were independently associated with the PSG measures-identified OSA condition. Considering that the clinical characteristics and metabolomic signatures of patients in the discovery and validation sets were similar, a merged dataset of clinical and metabolic features from the discovery and validation sets were used in the following regression models. As shown in Figure 3B, the results demonstrate that most metabolites still remained in significant association with OSA even after adjusting for BMI, neck circumference, waist circumference, hip circumference, waist-to-hip ratio, night-time SBP and DBP, TG, and HDL-C. These results indicate that OSA might cause specific metabolite changes in hypertensive patients.

### 3.4. Metabolic Pathway Enrichment and Functional Analysis

To characterize the key metabolic pathways involved in the differentiated metabolites that discriminated patients with OSA and non-OSA patients, pathway analysis was performed utilizing MetaboAnalyst. As shown in Figure 4A, a variety of metabolic pathways were significantly enriched, mainly including multiple amino acid metabolism and lipid metabolism (e.g., glutamate metabolism, taurine metabolism, glutathione metabolism, bile acid biosynthesis, phospholipid biosynthesis, and arachidonic acid metabolism). To further understand the biological function and latent diseases of significant metabolites, a functional relationship network analysis was performed using the Ingenuity Pathway Analysis (IPA) knowledge database. The resultant network revealed that those metabolites were primarily involved in the processes of inflammation, oxidative stress, and lipid peroxidation, and showed significant association with hypertension, cardiovascular diseases, hypoxia, and inflammatory diseases (Figure 4B).

### 3.5. Metabolites Association with PSG Measures, Blood Pressure, and Cardiovascular Risk Factors

The associations between the altered serum metabolites and the clinical/laboratory observations were performed by using Spearman’s rank correlation coefficients. As expected, most of the metabolites that distinguished hypertensive patients with OSA from those without OSA were strongly associated with PSG parameters, including AHI, CT90, ODI, LSaO_2_, MSaO_2_, ArI, and MAD (*p* < 0.05). LPC species, DAG species, fatty acyls, phosphatidylcholine, indoles, and eicosanoids were associated positively, whereas most of amino acid and its derivatives were associated negatively (Figure 5A). Regarding BP measures and the commonly known cardiovascular risk lipid factors, their correlations with metabolite signatures were slight or moderate (Figure 5A). Several LPC and eicosanoid species (i.e., 5-HETE, LPC 18:0) were found to be positively associated with night time-SBP and night time-DBP, whereas taurine, methionine, and histidine were associated negatively. In addition, the correlogram also revealed that a variety of lipids and amino acid and its derivatives were associated with TG and HDL-C. Notably, similar associations between metabolites and clinical/laboratory measures were also observed in the major debiased sparse partial correlation (DSPC) network analyses network (Figure 5B). The resultant network also demonstrated that these altered metabolites showed strong interactions with each other, indicating a synergistic effect on the clinical phenotypic changes.

### 3.6. Multi-Metabolites Model for Diagnosis of OSA in Hypertensive Individuals

Given that the large panel of differentially expressed metabolites that distinguished hypertension comorbid OSA from OSA-free patients (Appendix A and Figure 3B), we performed three machine-learning-algorithm-based feature selecting methods (PLS-DA, SVM, and random forest) to reduce the number of metabolic features. By using the discovery dataset, seven metabolites overlapping in the panels of the top twenty features from three machine learning methods were identified as potential markers for discriminating patients with OSA from those without OSA (Figure 6A). Six chemical standard-annotated metabolites were selected for subsequent model construction, including 5-hydroxyeicosatetraenoic acid (5-HETE), taurine, L-histidine, lysophosphatidic acid 16:0 (LPA 16:0), LPC 18:0, and dihydrosphingosine.

The univariate-receiver-operating curve (ROC) of each metabolite yielded an area under ROC (AUC) >0.8 for discriminating OSA from non-OSA subjects in both discovery and validation sets (Figure 6B,C). In addition, the logistic regression-based ROC models using the selected six features demonstrated a greater classification accuracy for OSA versus non-OSA (discover set: AUC = 0.995, CI = 0.991–1; validation set: AUC = 0.971; CI = 0.951–0.993; Figure 6B,C). The predictive performance of the six-metabolite diagnostic model was further evaluated by posterior classification probability. The resultant plot indicated that the established model could correctly classify 97.6% of patients with OSA and 99.2% of patients without OSA in discovery set (Figure 6D). The classification performance in the validation set was similar to that observed in the discovery set, highlighting a 96.8% correct classification for OSA subjects and 99.1% for non-OSA subjects (Figure 6E). The predictive accuracy of six-metabolite model in the discovery and validation set was also validated by the 500-times random permutation test with a significant *p* value < 0.0001 (Figure 6F,G).

## 4. Discussion

This work reports the first study using broad-spectrum metabolomic profiling in serum samples from a large series of hypertensive patients with and without OSA. The main findings of our study are the following: First, we demonstrated that OSA condition could cause marked and widespread changes in metabolomic phenotype of hypertensive patients, leading to a specific metabolite fingerprint consisting of a variety of hydrophilic amino acids, phospholipids, and arachidonic acid derivative eicosanoids. Second, we described that these metabolic alterations synergistically interacted with various metabolic pathways and biological signals and are closely associated with clinical characteristics of hypertension comorbid OSA. Finally, we developed a multi-metabolite model comprising six metabolic biomarkers that could precisely discriminate OSA from non-OSA in hypertensive individuals.

There is growing evidence that oxidative stress plays an important role in the pathophysiology of hypertension and associated cardiovascular consequences [29,30]. Oxidative stress has also been widely reported to be closely associated with OSA due to the chronic intermittent hypoxia [31]. Notably, our metabolic data demonstrated a generalized suppression of endogenous antioxidants in the serum samples of hypertensive patients with OSA compared to non-OSA patients (Figure 4B). Taurine is a naturally occurring sulfur-containing amino acid that is expressed in the majority of tissues [32]. It exerts antioxidant, anti-inflammatory, and anti-hyperlipidemic effects and has shown benefits to the cardiovascular system [32,33,34]. Histidine is an essential amino acid that has been associated with antioxidant effects, which is mediated by the scavenging of reactive oxygen and nitrogen species [35]. Previous evidence also revealed that circulating glutamine might play antioxidant and anti-inflammatory roles in cardiovascular pathophysiology by inducing the expression of heme oxygenase-1, heat shock proteins, and glutathione [36]. Taken together, these declined circulating metabolic antioxidants in hypertensive patients with OSA closely reflect a more severe oxidative stress condition within hypertension comorbid OSA subjects comparing to non-OSA hypertensive patients.

Although the pathogenesis of cardiovascular complications by OSA is not fully understood, several studies have suggested that OSA could promote systemic inflammation, leading to an increased BP and cardiovascular risk observed in these patients [37,38]. Here, our metabolic analyses also revealed hypertensive patients with OSA showed significantly altered lipid species implicating in several inflammatory signaling pathway (Figure 4). We observed that dihydrosphingosine, arachidonic acid, and its derivative eicosanoids, lysophosphatidic acid 16:0 (LPA 16:0), and a panel of LPC species (e.g., LPC 16:0, LPC 18:0) were increased in the serum of hypertensive patients with OSA compared to those without OSA (Figure 2 and Appendix A). Interestingly, these altered lipid markers were also observed in OSA subjects versus healthy controls [39,40,41]. Phospholipase A2 plays crucial roles in the release of arachidonic acid, LPA, and LPC from phospholipids. Its activation has been reported to be positively associated with chronic intermittent hypoxia under OSA condition [40,42]. Numerous evidence suggested that LPA, LPC, arachidonic acid, and its derivative eicosanoids (e.g., 5-HETE, 12-HETE) are important pro-inflammatory and pro-atherosclerotic lipid mediators in promoting the development of atherosclerotic cardiovascular diseases via multiple pathophysiological processes [43,44,45,46].

As discussed above, the current metabolomic data strongly demonstrated remarkable negative metabolic and cardiovascular effects of OSA on hypertensive subjects. However, OSA still remained a significantly undiagnosed and untreated condition in the majority of individuals, especially of patients with hypertension [3,11,47]. The current gold-standard for diagnosing OSA is overnight PSG, which is a complex, laborious, slow, and costly procedure, often inconvenient for patients and PSG operators [48,49]. Therefore, developing easily testable and reliable biomarkers for diagnosing OSA has been the focus of research for more than a decade [17,49,50].

Here, we identified a panel of serum metabolites which were independently associated with OSA condition in hypertensive patients (as shown in Figure 3B). Furthermore, these potential metabolic biomarkers showed significant correlation with PSG markers, including AHI, ArI, MAD, LSaO_2_, MSaO_2_, CT90, and ODI (Figure 5). Using an advanced machine learning algorithm-based feature reduction method [26,28,51,52], a small panel of six metabolites were selected as biomarkers for the discrimination of hypertensive individuals with and without OSA. Of note, the AUC values of the six metabolite-based diagnostic model were up to one for the discrimination of OSA and non-OSA patients in both the discovery and validation sets (Figure 6), which was quite high and fully proved the significant value of metabolites in the diagnosis of OSA.

Our study has some limitations. Firstly, the participants of this study were Chinese people, which might limit the generalizability of our findings to other populations. Secondly, it is an observational study, and more experimental data is needed to identify the pathophysiological mechanism of altered metabolites. Thirdly, it is a single-center study, the utility of metabolite biomarkers for diagnosing OSA in an expanded multi-center hypertensive cohort is needed in future studies. Finally, further study focusing on these metabolic markers in predicting cardiovascular risk in hypertensive patients is greatly needed.

## 5. Conclusions

In summary, by characterizing the serum metabolome and lipidome landscape in a large-scale number of hypertensive patients with and without OSA (*n* = 559 in total), the present study offers a comprehensive view of metabolic alterations in hypertension comorbid OSA and identifies a panel of metabolite biomarkers that could effectively identify OSA from the hypertensive population, providing new mechanistic avenues to better understand the fundamental biological processes and advance the diagnostic approaches in the related domain. Since these altered metabolites are related to a variety of biochemical pathways that are implicated in the development of cardiovascular diseases, our study highlights that the importance of early diagnosis and treatment of OSA is crucial for patients with hypertension comorbid OSA.

## Figures and Tables

**Figure 1 antioxidants-11-01946-f001:**
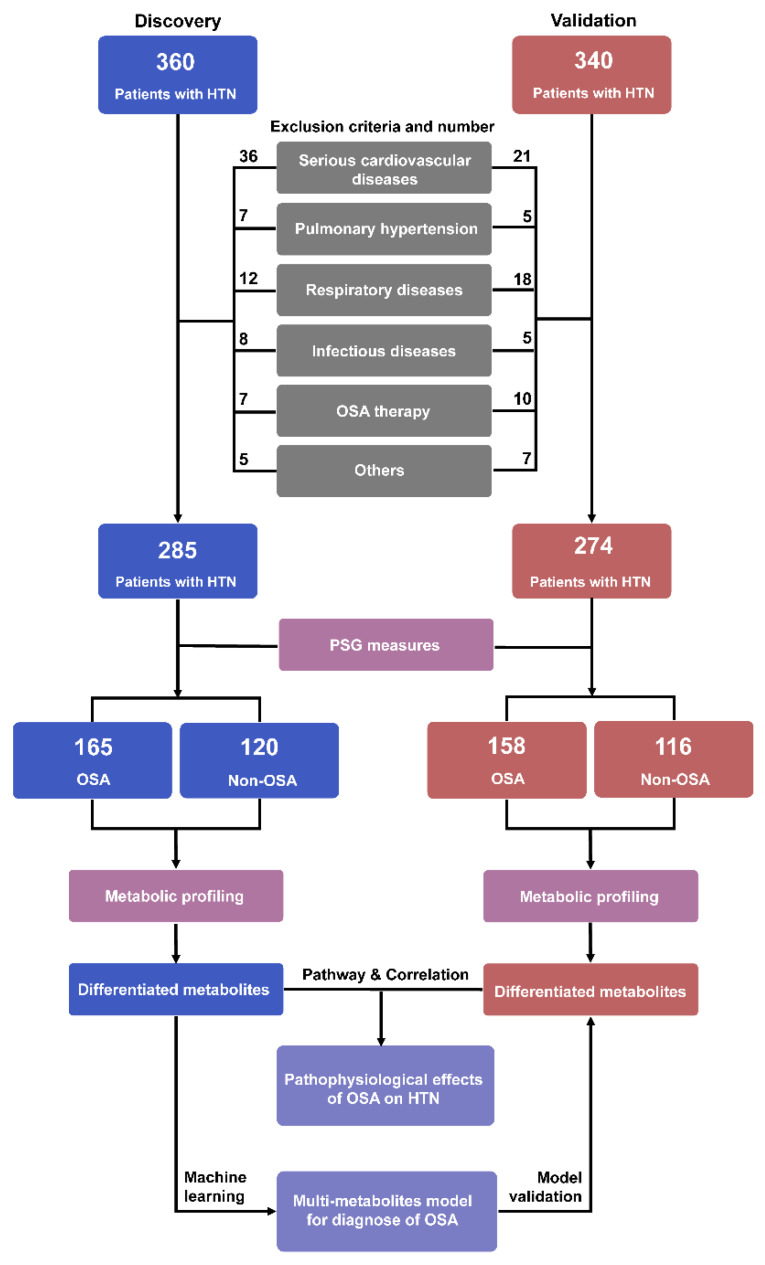
Flowchart of sample selection and study design for the study. HTN, hypertension; OSA, obstructive sleep apnea.

**Figure 2 antioxidants-11-01946-f002:**
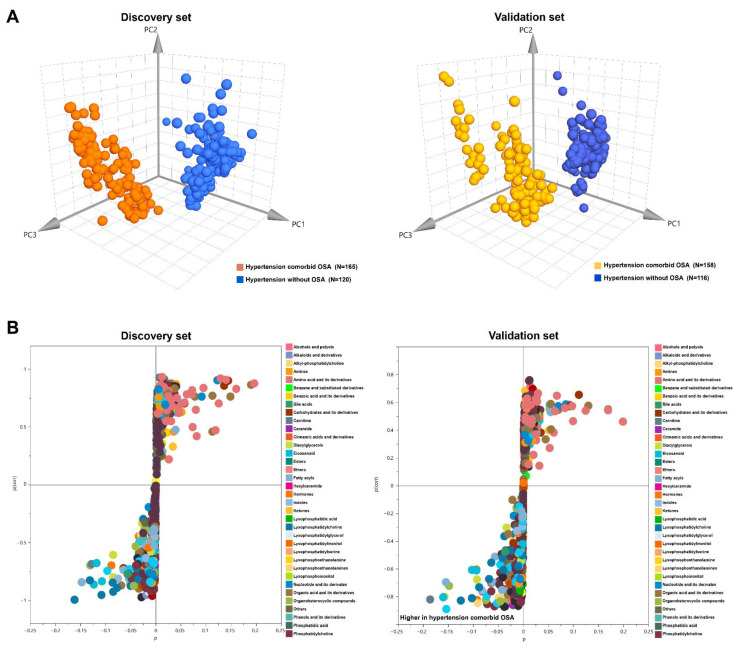
Multivariate statistical analysis of metabolic profiling data from discovery and validation sets. (**A**) Principal component analysis score plots showing apparent separation of patients with hypertension comorbid OSA from hypertensive patients without OSA. Each point represents an individual serum sample. (**B**) *S*−plot of orthogonal partial least squares discriminate analysis depicting remarkable metabolic signatures between hypertensive patients with OSA and those without OSA. Upper right zone: metabolites that are more abundant in hypertension without OSA in comparison to hypertension comorbid OSA. Lower left zone: metabolites that are more abundant in hypertension comorbid OSA in comparison to hypertension without OSA. Color coding on the left represents the different metabolite families.

**Figure 3 antioxidants-11-01946-f003:**
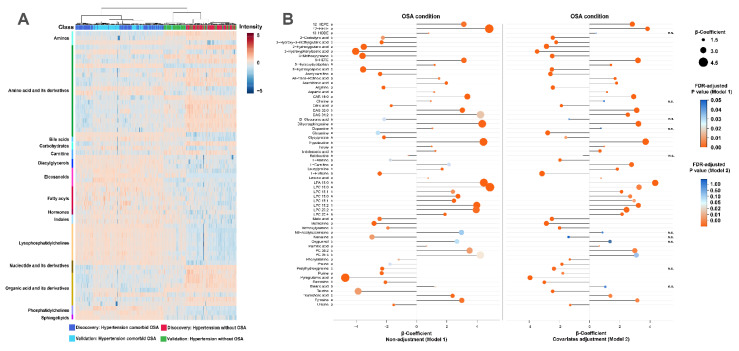
Metabolic changes associated with hypertension comorbid OSA. (**A**) Heat map of the datasets of differential metabolites, indicating that the expressed trends of most differential metabolites that distinguished patients with hypertension comorbid OSA from non-OSA hypertensive patients were similar in the discovery and validation sets. (**B**) Regression analysis plot depicting the association of each metabolite and OSA condition; positive or negative β-coefficient values indicate a positive or negative correlation between metabolites and OSA condition, FDR-adjusted *p* < 0.05 was considered as significant threshold, n.s. represents no significance. a and b indicates if the metabolites are annotated by chemical-standards or not. Model 1: Univariate regression analysis; Model 2: Adjustments for changed clinical variables and commonly known metabolic factors, including ages, sex, BMI, neck circumference, waist circumference, hip circumference, waist-to-hip ratio, night-time SBP, night-time DBP, TC, HDL-C. C Pathway enrichment analysis of the differentiated metabolites by using MetaboAnalyst.

**Figure 4 antioxidants-11-01946-f004:**
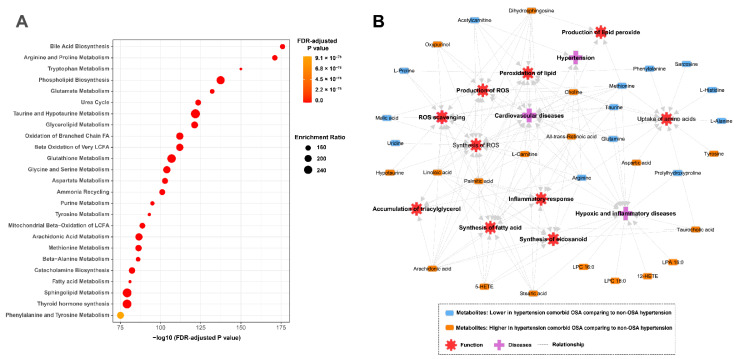
Pathway and function analyses of differentiated metabolites. (**A**) Metabolic pathway enrichment analysis by using MetaboAnalyst. The bubble size refers to the enrichment factor of the pathway and the color represents the FDR-adjusted *p* value of pathway significance. (**B**) Functional network between the metabolic signatures and biological pathways/diseases by using Ingenuity Pathway Analysis Database.

**Figure 5 antioxidants-11-01946-f005:**
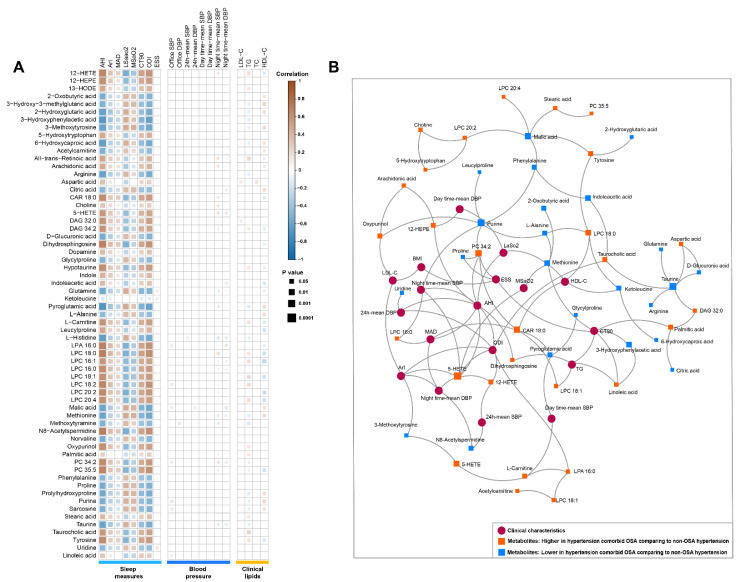
Correlation network of metabolic signatures and cli87nical variables. (**A**) Spearman’s rank correlation plot depicting the association of each differentiated metabolite with sleep measures, blood pressures, and clinical lipids. Positive correlations are displayed in orange and negative correlations in blue. Color intensity and size of the square are proportional to the Spearman correlation coefficients. Only metabolites with *p* < 0.05 are depicted with square labels. (**B**) Debiased sparse partial correlation network plot visualizing the major relationships of key metabolites and clinical variables; squares represent metabolites, circles represent clinical variables, the solid lines indicate significant correlation with a |coefficients| > 0.5.

**Figure 6 antioxidants-11-01946-f006:**
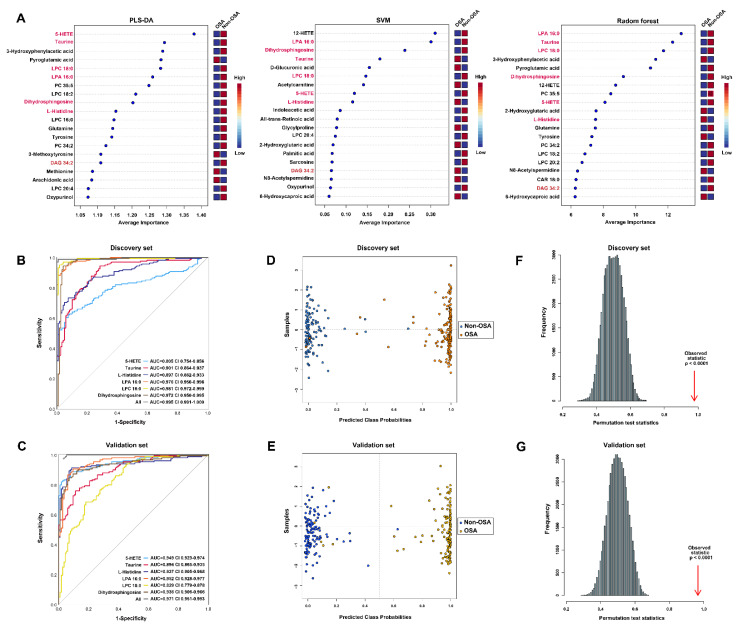
Multi-metabolite model for diagnosing OSA in hypertension by using multiple machine learning algorithms and receiver operating curve. (**A**) The top 20 metabolites were ranked on the basis of their average importance contributing to the diagnosis of OSA in hypertension (classification accuracy). The overlapping six metabolites obtained from three machine learning algorithms were marked in red. (**B**,**C**) Logistic regression-based receiver-operating characteristic curve analyses of the diagnostic performances for individual metabolite and their combinations in discovery and validation set. (**D**,**E**) Plot of posterior classification probability showing the segregation of hypertension comorbid OSA and non-OSA hypertension samples according to the predictive six-metabolite model. Each symbol represents the classification probability that a given sample belongs to OSA or to non-OSA group. The classification boundary is shown by a dotted line at x = 0.5. (**F**,**G**) The permutation test (*n* = 1000 times) showing the robustness of predictive model generated by six-metabolites.

**Table 1 antioxidants-11-01946-t001:** Demographic and clinical characteristics of hypertensive subjects.

	Discovery Set			Validation Set		
Variables	OSA (*n* = 165)	Non-OSA (*n* = 120)	*p*	OSA (*n* = 158)	Non-OSA (*n* = 116)	*p*
General characteristics						
Ages	46.2 ± 13.7	43.2 ± 12.4	0.059	45.67 ± 12.38	45.28 ± 11.73	0.8
Male, *n* (%)	120, (74.53%)	86, (71.67)	0.59	115, (77.18%)	81, (69.83%)	0.18
Current smoker, *n* (%)	108, (67.08%)	77, (64.17%)	0.61	100, (67.11%)	73, (63.79%)	0.57
Alcohol intake > 10 g/day, *n* (%)	99, (61.49%)	71, (59.71%)	0.69	97, (65.10%)	72, (62.07%)	0.61
Diabetes mellitus, *n* (%)	43, (26.88%)	33, (27.50%)	0.91	44, (27.85%)	29, (25.00%)	0.60
Sleeping pills users, *n* (%)	8, (4.85%)	5, (4.17%)	0.79	11, (6.96%)	4, (3.46%)	0.21
Neck circumference (cm)	42.46 ± 4.10	40.64 ± 4.13	0.006	42.47 ± 3.97	40.95 ± 3.95	0.034
Waist circumference (cm)	103.35 ± 11.43	95.52 ± 13.02	<0.0001	102.93 ± 10.38	97.01 ± 10.44	0.0004
Hip circumference (cm)	108.40 ± 8.96	104.65 ± 11.90	0.026	107.89 ± 10.13	103.63 ± 8.44	0.013
Waist-to-Hip ratio	0.96 ± 0.067	0.92 ± 0.069	0.0002	0.96 ± 0.052	0.93 ± 0.043	0.0043
BMI (kg/m^2^)	29.93 ± 4.48	27.26 ± 4.11	<0.0001	29.47 ± 4.67	27.52 ± 4.92	0.0016
Heart rate (times/min)	73.03 ± 9.47	72.30 ± 8.92	0.54	72.10 ± 9.64	73.86 ± 9.84	0.17
BP parameters						
Mean office SBP (mmHg)	141.65 ± 20.25	137.78 ± 18.08	0.11	144.31 ± 19.90	142.29 ± 19.24	0.44
Mean office DBP (mmHg)	91.58 ± 15.79	90.42 ± 12.95	0.53	92.89 ± 15.59	91.85 ± 15.09	0.61
24-h SBP (mmHg)	134.05 ± 16.55	130.20 ± 13.05	0.069	134.80 ± 14.13	131.71 ± 14.90	0.14
24-h DBP (mmHg)	85.83 ± 11.79	84.72 ± 9.86	0.47	87.05 ± 10.07	84.57 ± 10.99	0.11
Day-time SBP (mmHg)	137.43 ± 16.54	134.27 ± 13.36	0.14	137.95 ± 14.39	133.84 ± 15.17	0.057
Day-time DBP (mmHg)	88.03 ± 11.82	87.76 ± 9.60	0.86	88.86 ± 9.78	86.08 ± 10.59	0.061
Night-time SBP (mmHg)	128.50 ± 18.86	123.51 ± 16.20	0.045	130.98 ± 16.98	126.06 ± 16.04	0.044
Night-time DBP (mmHg)	82.05 ± 13.84	78.54 ± 10.41	0.042	84.43 ± 12.32	80.11 ± 11.33	0.014
Non-dipper, *n* (%)	75, (45.45%)	46, (38.33%)	0.23	77, (48.73%)	48, (41.38%)	0.23
Laboratory tests						
Fasting blood glucose (mmol/L)	6.05 ± 1.39	5.79 ± 1.41	0.14	6.27 ± 2.14	5.91 ± 1.66	0.17
LDL-C (mmol/L)	3.02 ± 0.98	3.08 ± 0.85	0.65	3.06 ± 0.83	3.05 ± 0.84	0.94
TC (mmol/L)	4.93 ± 1.05	4.87 ± 1.08	0.65	4.95 ± 0.99	4.90 ± 0.89	0.71
TG (mmol/L)	1.78 [1.39, 2.49]	1.43 [1.07, 2.27]	0.002	1.86 [1.21, 2.83]	1.53 [1.09, 2.25]	0.046
HDL-C (mmol/L)	1.11 ± 0.24	1.25 ± 0.26	<0.0001	1.17 ± 0.24	1.26 ± 0.38	0.043
ALT (mmol/L)	28.98 ± 14.41	28.25 ± 15.81	0.77	30.56 ± 13.03	28.87 ± 14.38	0.55
Non-HDL (mmol/L)	3.74 ± 1.13	3.61 ± 1.02	0.35	3.57 ± 1.29	3.42 ± 1.25	0.34
AST (mmol/L)	26.39 ± 10.25	25.05 ± 9.26	0.39	26.81 ± 12.45	24.48 ± 11.48	0.33
Sleep parameters						
AHI(events/h)	39.66 ± 22.41	6.42 ± 5.29	<0.0001	35.94 ± 18.77	8.63 ± 3.85	<0.0001
LaSO_2_(%)	78.25 ± 7.79	87.94 ± 5.62	<0.0001	76.82 ± 10.84	84.88 ± 10.91	<0.0001
MSaO_2_(%)	93.58 ± 2.16	95.61 ± 1.57	<0.0001	93.81 ± 2.09	94.90 ± 1.62	<0.0001
MAD(s)	26.19 ± 5.66	22.02 ± 8.01	0.00015	25.21 ± 6.08	22.72 ± 6.66	0.035
CT90(%)	4.12 [1.36, 12.80]	0.10 [0.00, 0.31]	<0.0001	5.70 [1.74, 16.29]	0.20 [0.00, 1.00]	<0.0001
ODI(events/h)	31.81 [19.98, 47.46]	4.65 [1.14, 8.75]	<0.0001	30.90 [19.34, 43.95]	8.65 [4.84, 11.86]	<0.0001
ArI(events/h)	20.85 [10.81, 34.13]	8.60 [4.51, 12.93]	<0.0001	18.10 [5.03, 32.35]	10.40 [4.08, 27.90]	0.00096
ESS score	8.75 ± 5.06	7.28 ± 5.12	0.018	9.49 ± 5.01	8.04 ± 4.88	0.022

Continuous data are presented as mean ± standard deviation or median [interquartile range], categorical variables are presented as %. A two-tailed Student’s *t* test or Mann Whitney *U* test were used for continuous data in the comparison of OSA and non-OSA. The Chi-square test was used for categorical data. BMI, body mass index; BP, blood pressure; AHI, apnea-hypopnea index; LSaO_2_, lowest oxygen saturation; MSaO_2_, mean oxygen saturation; MAD, mean apnea-hypopnea duration; CT90, percentage of cumulative time with oxygen saturation below 90%; ODI, oxygen desaturation index; ArI, arousal index; ESS, Epworth sleepiness scale; ALT, alanine aminotransferase; AST, aspartate aminotransferase; LDL-C, low-density lipoprotein cholesterol; TC, total cholesterol; TG, triglycerides; HDL-C, high-density lipoprotein cholesterol; non-HDL, non-high-density lipoprotein cholesterol.

## Data Availability

The data are contained within the article and the Appendix A.

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
