# Peer review of "Comprehensive Metabolomics and Machine Learning Identify Profound Oxidative Stress and Inflammation Signatures in Hypertensive Patients with Obstructive Sleep Apnea"

_antioxidants, 2022, doi:10.3390/antiox11101946_

Round 1
Reviewer 1 Report
Summary:
Authors studied 559 hypertensive patients with and without sleep apnea (OSA). They performed metabolome and lipidome-wide analyses to highlight potential biomarkers for diagnosing OSA in this population. They proposed unique sera metabolic phenotype characterized by abnormalities in oxidative stress and inflammation. Using three machine learning algorithms, they constructed six discriminatory metabolites to define diagnostic and classified model. Notably, the established multivariate-model could accurately identify OSA subjects. They propose metabolomics to be tool for improvement of the diagnosis of OSA in hypertension.
Major comment(s):
1. (Night blood pressure) dippers x non-dippers should be analyzed - might have importance for definition of sleep apnea in hypertensive patients. Was there any difference between OSA vs. non-OSA?
2. Smoking status and especially drinking status (what does it mean “alcohol” in Table 1?) should be better defined/described. In addition, use of sleeping pills could be described or at least discussed.
3. If authors measured neck circumference - also waist circumference should be measured - as a marker of central obesity it could be associated with metabolome/lipidome as well. They described BMI and BP as metabolic regulators, waist circumference, waist to hip ratio … could be even stronger metabolic players. Were there any patients suffering from diabetes mellitus?
Minor comments/suggestions:
1. Non-HDL cholesterol should be also analyzed
2. Were there any sex/gender differences observed?
3. Duration and (effectiveness/type of) treatment could be listed/discussed - multiple use of drugs, diuretics, beta-blockers with potential of metabolic effect?
4. What does it mean “Official” BP (Table 1)- measured in office, causal … ?
5. Some expressions could be improve (“evidencing” – page 6/r 402)
In summary:
Very complex study of potential metabolic indicators (expressed as metabolome) in hypertensive patients with sleep apnea is described in reasonably numerous population of hypertensive. The study is based on very sophisticated laboratory and statistical approaches and is well written. Some basic parameters are to be added/described or at least discussed, which could have also profound influence on the findings (central obesity, compensation of diabetes mellitus, …). Please see my comments.
Author Response
Dear Reviewer,
We would like to submit a revised manuscript entitled “Comprehensive metabolomics and machine learning identify profound oxidative stress and inflammation signatures in hypertensive patients with obstructive sleep apnea (ID: antioxidants-1916426)” by Z.Y. Du et al. for your consideration for publication as a research article in Antioxidants. We have revised our manuscript according to the valuable comments of reviewers. All the changes were marked up using the “Track Changes” in the main text, and the responses are listed point by point in the section of Reply to Reviewers of the resubmission window.
All the co-authors have approved the revised version of this manuscript and we hope that the revised manuscript will be worthy of publication in Antioxidants.
Major comment 1: (Night blood pressure) dippers x non-dippers should be analyzed - might have importance for definition of sleep apnea in hypertensive patients. Was there any difference between OSA vs. non-OSA?
Response 1: Thank you for your critical and thoughtful comments. Your comments are very useful for improving the quality of our article. As per your suggestion, the results of dippers x non-dippers in OSA vs. non-OSA has been given in Table 1 and described in the section of 3.1. Participant clinical characteristics in the revised manuscript. Although no significant difference was observed in the prevalence of non‑dipping pattern in the comparisons of OSA and non-OSA groups, the results demonstrated that hypertensive patients with OSA (45.45% in discovery set; 48.73% in validation set) exhibited a higher trend in the prevalence of non‑dipper than patients without OSA (38.33% in discovery set; 41.38% in validation set). Considering the small sample size of this study, we believed that a future large-scaled study of hypertensive subjects with and without OSA condition may give a more accurate result for the prevalence of night-BP dipper.
Major comment 2: Smoking status and especially drinking status (what does it mean “alcohol” in Table 1?) should be better defined/described. In addition, use of sleeping pills could be described or at least discussed.
Response 2: Thanks a lot for your critical comments. We are very sorry for our insufficient description in the definition of smoking status and drinking status. In the present study, smoking status was defined as current smokers at the time of health checkup examination. Drinking status was defined as subjects with an alcohol consumption >10g/day. The criteria used to define smoking status and drinking status has been given in the section of Data collection and clinical laboratory tests in the revised manuscript. As per your suggestion, the datasets of use of sleeping pills have been depicted in Table 1, and the corresponding results revealed that the proportion of sleeping pills users in patients with OSA were 4.85% and 6.96% in the discovery set and validation set, the proportion in patients without OSA were 4.17% and 3.46%, respectively.
Major comment 3: If authors measured neck circumference - also waist circumference should be measured - as a marker of central obesity it could be associated with metabolome/lipidome as well. They described BMI and BP as metabolic regulators, waist circumference, waist to hip ratio … could be even stronger metabolic players. Were there any patients suffering from diabetes mellitus?
Response 3: Thanks a lot for your thoughtful comments. Your comments are very useful for improving our work. According to your comments, we have provided the results of waist circumference, hip circumference, waist-to-Hip ratio, and the prevalence of diabetes mellitus in the Table 1 of the revised manuscript. Our results indicated that hypertensive patients with OSA had higher values of waist circumference, hip circumference, and waist-to-Hip ratio than patients without OSA. In discovery and validation set, the prevalences of diabetes mellitus in patients with OSA were 26.88% and 27.55%, and the prevalences in patients without OSA were 27.50% and 25.00%, respectively. There were no significant differences in the prevalence of diabetes mellitus between OSA and non-OSA. As per your suggestions, the waist circumference, hip circumference, waist-to-Hip ratio, and diabetes mellitus were selected as the adjusted-variables (potential metabolic regulators) in the multivariate regression analysis in the revised manuscript. The corresponding results have been depicted in the revised Figure 3B.
Minor comment/suggestion 1: Non-HDL cholesterol should be also analyzed.
Response 1: Thanks a lot for your useful comments. The results of non-HDL cholesterol have been given in the Table 1 of the revised manuscript.
Minor comment/suggestion 2: Were there any sex/gender differences observed?
Response 2: Thanks a lot for your thoughtful comments. In general, sex/gender may be considered as potential metabolic regulation factor. Therefore, we employed unsupervised and supervised multivariate statistical analyses to test whether the gender could give a potential impact on the identified metabolite markers that distinguished OSA from non OSA. As shown in Figure S2 of the revised Supplementary material, unsupervised principal component analysis (PCA) and supervised partial least square discriminant analysis (OPLS-DA) did not show clustering of samples by gender, indicating no discriminatory metabolite signatures due to gender in the study groups.
Minor comment/suggestion 3: Duration and (effectiveness/type of) treatment could be listed/discussed - multiple use of drugs, diuretics, beta-blockers with potential of metabolic effect?
Response 3: Thanks a lot for your thoughtful comments. Your comments are very useful for us. As per your suggestion, the detailed use of antihypertensive drugs has been given in Figure S3 of the revised Supplementary material. In addition, we performed unsupervised PCA and supervised partial least square discriminant analysis (PLS-DA) to investigate whether different drug or drugs combinations could influence the profiles of the discriminatory metabolite signatures. As shown in Figure S3, the score plots of unsupervised and supervised multivariate statistical analyses of discriminatory metabolite datasets did not show clustering of samples by different drug or drug combinations, suggesting that different drugs might not give a remarkable metabolic effects on these metabolite biomarkers in distinguishing OSA from non OSA.
Minor comment/suggestion 4: What does it mean “Official” BP (Table 1)- measured in office, causal … ?
Response 4: Thanks a lot for your critical comments. We feel sorry that our language issues made you feel confused. The word “Official” has been replaced as “Office”. The mean office SBP and DBP of the studied individuals in Table 1 was obtained from three consecutive measurements at 5-minute intervals.
Minor comment/suggestion 5: Some expressions could be improve (“evidencing” – page 6/r 402)
Response 5: Thanks a lot for your critical comments. We feel sorry very sorry that our language error and presentation issues made you feel confused. We have revised the text to address your concerns and hope that it is now clearer.
In summary: Very complex study of potential metabolic indicators (expressed as metabolome) in hypertensive patients with sleep apnea is described in reasonably numerous population of hypertensive. The study is based on very sophisticated laboratory and statistical approaches and is well written. Some basic parameters are to be added/described or at least discussed, which could have also profound influence on the findings (central obesity, compensation of diabetes mellitus, …). Please see my comments.
Response: Dear reviewer, thanks a lot for your critical and thoughtful comments. Your kind comments are very useful for improving our article. All the responses to your comments have been listed point by point in the above section. All the related changes were marked up using the “Track Changes” in the revise manuscript and Supplementary material. Again, we fully thank for your time and attention in our manuscript, and we hope that the revised article can achieve your expectation and will be worthy of publication in Antioxidants.
Reviewer 2 Report
This is an interesting paper, well written with an interesting argument. I approve it for the publication.
the originality of the study is high if considering the difficult of predicting the risk of apnea-hypopnea in hypertensive individuals considering that pathophysiological process is still incompletely understood.The use of machine learning tool to explore the pathophysiological processes of hypertension comorbid OSA considering the Metabolome and lipidome-wide analyses to derive potential biomarkers for diagnosing OSA in hypertensive subjects is a scientific soundness.
Author Response
Point 1: This is an interesting paper, well written with an interesting argument. I approve it for the publication. The originality of the study is high if considering the difficult of predicting the risk of apnea-hypopnea in hypertensive individuals considering that pathophysiological process is still incompletely understood. The use of machine learning tool to explore the pathophysiological processes of hypertension comorbid OSA considering the metabolome and lipidome-wide analyses to derive potential biomarkers for diagnosing OSA in hypertensive subjects is a scientific soundness.
Response 1: Dear reviewer, thanks a lot for your time and attention in our manuscript. We feel very grateful for your recognition of this work. We hope that the revised manuscript will be worthy of publication in Antioxidants.
Reviewer 3 Report
- The figures are too small to be seen and hard to follow with their explanations in the texts. Suggest that more prominent figures, especially for Figure 3, could be uploaded separately with high solutions.
- The authors enrolled two groups of patients for the study, the discovery set and the validation set. However, most examinations could be conducted for all patients as they had similar significant results shown in two datasets separately. In Figure 1, under ‘shared metabolites,’ it was confusing to test names listed separately for two datasets. It was correct to have identified 6 metabolites for discriminating OSA patients from hypertension patients in the discovery set, then tested the discriminant ability of the metabolites in the validation set (as shown in Figure 5). It would be better for the authors to mention which and why tests were conducted for separate or combined groups of patients and highlight different findings if identified by separate groups.
Below are some minor comments:
1. Line 80, page 2, ‘the discovery set’ should be ‘validation set?’
2. Excluded number of diseases should be given in Figure 1; alternatively, indicated in section 2.1.
3. Abbreviations in Figure 1 should be shown in the legend.
4. Suggest that some changes for variable names be more precise, e.g., ‘Alcohol,’ ’24 BP’; and measurement unit, e.g., ‘kg/m2’ in Table 1. The full name of ESS should be provided in the undernotes.
Author Response
Dear Reviewer,
We would like to submit a revised manuscript entitled “Comprehensive metabolomics and machine learning identify profound oxidative stress and inflammation signatures in hypertensive patients with obstructive sleep apnea (ID: antioxidants-1916426)” by Z.Y. Du et al. for your consideration for publication as a research article in Antioxidants. We have revised our manuscript according to the valuable comments of reviewers. All the changes were marked up using the “Track Changes” in the main text, and the responses are listed point by point in the section of Reply to Reviewers of the resubmission window.
All the co-authors have approved the revised version of this manuscript and we hope that the revised manuscript will be worthy of publication in Antioxidants.
Point 1: The figures are too small to be seen and hard to follow with their explanations in the texts. Suggest that more prominent figures, especially for Figure 3, could be uploaded separately with high solutions.
Response 1: Thank you very much for your critical comments. We are very sorry for the unsatisfactory figures in last submission that made you feel confused. According to your suggestions, we have separately uploaded high-solutions figures (TIFF and PDF) in the revised manuscript. Furthermore, Figure 3 has been split into two figures, namely Figure 3 and Figure 4. We hope that it is now clearer.
Point 2: The authors enrolled two groups of patients for the study, the discovery set and the validation set. However, most examinations could be conducted for all patients as they had similar significant results shown in two datasets separately. In Figure 1, under ‘shared metabolites,’ it was confusing to test names listed separately for two datasets. It was correct to have identified 6 metabolites for discriminating OSA patients from hypertension patients in the discovery set, then tested the discriminant ability of the metabolites in the validation set (as shown in Figure 5). It would be better for the authors to mention which and why tests were conducted for separate or combined groups of patients and highlight different findings if identified by separate groups.
Response 2: Thanks a lot for your critical and thoughtful comments. In the present study, patients in discovery set (Sleep center of Beijing Anzhen Hospital) were enrolled between March 2017 and June 2018, patients in the validation set (Cardiology outpatient department of of Beijing Anzhen Hospital) were enrolled from January 2019 to November 2020. Firstly, the metabolomic analysis were performed to investigate the potential biomarkers between OSA and non-OSA hypertensive patients in discovery set. Subsequently, the metabolomic data of validation set were separately collected to test whether the metabolomic features of patients in validation set were also analogous to those of patients in the discovery set. As you mentioned, the results of clinical characteristics and the identified metabolomic alterations in discovery and validation sets were similar. These results indicated that metabolomic phenotypes closely reflected the clinical phenotypes, and the identified metabolomic biomarkers were effective for distinguishing OSA and non-OSA in two sets of hypertensive subjects collecting from different time-point and outpatient clinic.
Because the expressed trends of the sixty-three discriminatory metabolites between OSA and non-OSA subjects were similar (as shown in Table S1 and Figure 3A) in the discovery and validation sets, the pathway enrolment analysis and the correlation analyses of metabolites with OSA condition/clinical features were performed by using the merged datasets of differential metabolites in both discovery and validation sets. In the following biomarker model construction, by using three machine learning algorithms, we firstly identified a small number of six representative biomarkers that can maintain a significant performance for OSA diagnosis in discovery set. To verify the discrimination performance of the six-metabolite discrimination model, the logistic model-based receiver operating curve (ROC), posterior classification probability, and permutation test were independently performed in the validation set. As shown in Figure 6 (Figure 5 in last version) of the revised manuscript, the resultant plot indicated that the established model could correctly classify 97.6% of patients with OSA in discovery set and 96.8% of patients with OSA in validation set. We have revised the Figure 1 and tests to address your concerns and hope that it is now clearer.
Minor comment 1: Line 80, page 2, ‘the discovery set’ should be ‘validation set?’
Response 1: Thanks a lot for your critical comments. We feel sorry that our incorrect writing made you feel confused. The “discovery set” has been replaced as “validation set”.
Minor comment 2:: Excluded number of diseases should be given in Figure 1; alternatively, indicated in section 2.1.
Response 2: Thanks a lot for your critical comments. As per your suggestion, excluded number of diseases has been given in Figure 1 of the revised manuscript.
Minor comment 3: Abbreviations in Figure 1 should be shown in the legend.
Response 3: Thanks a lot for your thoughtful comments. We are very sorry for our insufficient description in the legend of Figure 1. According to your suggestions, the abbreviations have been depicted in the legend of Figure 1 in the revised manuscript.
Minor comment 4: Suggest that some changes for variable names be more precise, e.g., ‘Alcohol,’ ’24 BP’; and measurement unit, e.g., ‘kg/m2’ in Table 1. The full name of ESS should be provided in the undernotes.
Response 4: Thanks a lot for your comments. Your comments are critical and useful for improving the quality of our article. According to your comments, the variable names and related measurement units of Table 1 have been carefully revised in the current version of manuscript. The full name of ESS has been provided in the undernotes.
Round 2
Reviewer 1 Report
All changes are to my opinion satisfactory.
Reviewer 3 Report
Thank you for taking the effort to revise this article. Great to share interesting findings.